# Differential Effects of Nonsteroidal Anti-Inflammatory Drugs in an In Vitro Model of Human Leaky Gut

**DOI:** 10.3390/cells12050728

**Published:** 2023-02-24

**Authors:** Michele d’Angelo, Laura Brandolini, Mariano Catanesi, Vanessa Castelli, Cristina Giorgio, Margherita Alfonsetti, Mara Tomassetti, Mara Zippoli, Elisabetta Benedetti, Maria Candida Cesta, Sandro Colagioia, Pasquale Cocchiaro, Annamaria Cimini, Marcello Allegretti

**Affiliations:** 1Dept. of Life, Health and Environmental Sciences, University of L’Aquila, 67100 L’Aquila, Italy; 2Dompé Farmaceutici S.p.A., via Campo di Pile snc, 67100 L’Aquila, Italy; 3Dompé Farmaceutici S.p.A., via De Amicis 95, 80131 Napoli, Italy; 4Sbarro Institute for Cancer Research and Molecular Medicine, Dept. of Biology, Temple University, Philadelphia, PA 19122, USA

**Keywords:** inflammation, oxidative stress, anti-inflammatory, macrophages, transepithelial electrical resistance, tight junction

## Abstract

The intestinal barrier is the main contributor to gut homeostasis. Perturbations of the intestinal epithelium or supporting factors can lead to the development of intestinal hyperpermeability, termed “leaky gut”. A leaky gut is characterized by loss of epithelial integrity and reduced function of the gut barrier, and is associated with prolonged use of Non-Steroidal Anti-Inflammatories. The harmful effect of NSAIDs on intestinal and gastric epithelial integrity is considered an adverse effect that is common to all drugs belonging to this class, and it is strictly dependent on NSAID properties to inhibit cyclo-oxygenase enzymes. However, different factors may affect the specific tolerability profile of different members of the same class. The present study aims to compare the effects of distinct classes of NSAIDs, such as ketoprofen (K), Ibuprofen (IBU), and their corresponding lysine (Lys) and, only for ibuprofen, arginine (Arg) salts, using an in vitro model of leaky gut. The results obtained showed inflammatory-induced oxidative stress responses, and related overloads of the ubiquitin-proteasome system (UPS) accompanied by protein oxidation and morphological changes to the intestinal barrier, many of these effects being counteracted by ketoprofen and ketoprofen lysin salt. In addition, this study reports for the first time a specific effect of R-Ketoprofen on the NFkB pathway that sheds new light on previously reported COX-independent effects, and that may account for the observed unexpected protective effect of K on stress-induced damage on the IEB.

## 1. Introduction

As evidence accumulates linking the health of the gut with a variety of acute and chronic illnesses, understanding the mechanisms underlying gut injury and protection is of paramount importance. The intestinal barrier is the main contributor to gut homeostasis, modulating the absorption of water, electrolytes, and nutrients from the lumen while restricting the passage of noxious luminal elements and microorganisms [1]. Alterations in the permeability of the intestinal epithelium disrupts gastrointestinal (GI) function, leading to damage with far-reaching consequences for the pathophysiology of not only GI tract disorders, but also autoimmune diseases, among others [2]. The front line of this barrier is preserved by only a single layer of specialized epithelial cells that are connected by tight junction (TJ) proteins. However, various factors support the barrier, among them mucins, antimicrobial molecules, immunoglobulins, and cytokines. Perturbations of the intestinal epithelium or supporting factors can lead to the development of intestinal hyperpermeability, termed “leaky gut” [3]. A leaky gut is characterized by loss of epithelial integrity and reduced function of the gut barrier and is associated with the absorption of pathogens or toxic and inflammatory substances in the bloodstream [3]. Intestinal permeability is an established feature of numerous inflammatory and autoimmune disorders impacting the digestive system, such as inflammatory bowel disease and celiac disease, for which intestinal permeability is considered a symptom, not a cause of disease. Other disorders have been indicated as possible long-term consequences of leaky gut. In fact, toxins from the intestine may release into the bloodstream and cause a chronic low-grade inflammatory response that may represent a risk factor in many diseases, including dysmetabolism, long-term disorders such as arthritis, chronic fatigue syndrome [4], chronic liver diseases [5], diabetes [6], multiple sclerosis [7], and even cognitive disorders [8].

The established causes of increased intestinal permeability include systematic erosion of the intestinal lining. The intestinal lining has numerous layers of defense and can sustain temporary injury because it is constructed to continually repair and replace itself. However, significant and recurring damage associated with chronic drug use, alcohol abuse, or radiation therapy may erode the barrier, leading to deep and penetrating ulcers. Ethanol (EtOH) exposure can affect intestinal permeability by modulating TJ proteins [9] and gut flora, leading to inflammation [10].

A leaky gut has also been associated with prolonged use of Non-Steroidal Anti-Inflammatory Drugs (NSAIDs) that may be toxic to the intestinal epithelium, causing erosions, perforations, and longitudinal ulcers in the gut [11,12]. Both acute and chronic ingestion of NSAIDs by healthy volunteers and patients have been reported to promote altered intestinal barrier dysfunction and hypermotility [13]. 

The harmful effect of NSAIDs on intestinal and gastric epithelial integrity is considered an adverse effect that is common to all drugs belonging to this class, and it is strictly dependent on NSAID properties to inhibit cyclo-oxygenase (COX) enzymes. However, factors such as drug distribution, potency, unrelated drug actions, and salt form can influence the specific tolerability profile of different members of the same class [3,14,15,16]. For instance, we previously compared the effects of ketoprofen L-lysine salt (KLS) and ketoprofen sodium salt in an in vitro model of gastric epithelium stressed with ethanol (EtOH) [17], and demonstrated a specific action of L-lysine (Lys) in counteracting the harmful effect of the drug on the gastric mucosa layers, due to the amino acid ability to scavenge 4-Hydroxynonenal (HNE)-protein adducts and balance oxidative stress levels [17].

The present study aims to compare the effects of distinct classes of NSAIDs, such as ketoprofen (K), Ibuprofen (IBU), and their corresponding lysine (Lys) and, only for IBU, arginine (Arg) salts, in an in vitro model of leaky gut with focus on the inflammatory-induced oxidative stress responses and related consequences on the ubiquitin-proteasome system (UPS), in order to evaluate differential effects in the epithelial damage recovery phase.

## 2. Materials and Methods

### 2.1. Experimental Model: Caco-2 EtOH-Stressed

To perform analyses on an intestinal epithelium in vitro model, the Caco-2 cell line, purchased from American Type Culture Collection (ATCC, HTB-37), was used. Caco-2 are colorectal carcinoma cells. Caco-2 cells were cultured in Dulbecco’s modified Eagle’s medium (DMEM) containing of 10% fetal calf serum (FCS), 100 U/mL penicillin, 2 mM glutamine (all from Corning, New York, NY, USA), and 1% non-essential amino acid solution (MEM) (Gibco, New York, NY, USA), that was changed every 2 to 3 days. Cells were passaged after partial digestion with 0.25% trypsin-EDTA (Corning, New York, NY, USA) and were seeded at the density of 3 × 10^5^ cells/cm^2^ (passages 3–9). In vitro experiments were performed for 20 days (20 Day in vitro, DIV) apart from the MTS assay because the metabolism of the control (CTR) cells does not allow for reading of the formazan produced (OVER signal, see paragraph relative to MTS assay for further details). Once the intestinal epithelial model was obtained, cells were exposed to EtOH to induce leaky gut conditions. In this regard, the epithelial model was pre-treated for 24 h with EtOH (concentrations tested include 2%, 4%, 6%, and 8%. The 6% concentration was chosen for the subsequent experiments) (Sigma, St. Louis, MO, USA). The medium containing EtOH was replaced with the medium containing NSAIDs, or control (culture medium w/o NSAIDs), and maintained in culture for 72 h. NSAID stock was prepared at the concentration of 50 mM in 1 ml of sterile water (Corning, New York, NY, USA). Regarding IBU and K, 20 µL of NaOH 5N were added. The final concentrations of the tested NSAIDs in the preliminary experiments were 0.25, 0.5, and 1 mM. For the subsequent experiments, we focused on 1 mM. 

Different sets of in vitro experiments were performed:CTREtOHEtOH + IBUEtOH + IBU-LysEtOH + IBU-ArgEtOH + KEtOH + KLSEtOH + ArgEtOH + Lys

### 2.2. Activated Macrophages Model

The murine macrophage cell line RAW 264.7 (ATCC TIB-71) was purchased from ATCC, and cells were cultured as indicated by the manufacturer. For the experiment, cells were seeded 10,000/cm^2^ for 24 h, starved for 4 h (w/o serum), and activated with lipopolysaccharides (LPS 10 ng/mL for 24 h; Sigma Aldrich, St. Louis, MO, USA). Then, activated macrophages were treated with 1 mM of the R or S enantiomer of K or KLS for 24 h to evaluate NFkB translocation (as reported in the Subcellular protein fractionation kit paragraph). 

### 2.3. Cell Viability Assay

Cells were seeded 3 × 10^5^ cells/cm^2^ in a 96 multiwell plate, and after 48 h, cells were exposed to different EtOH concentrations (2%, 4%, 6%, 8%) for 24 h, to find a sub-toxic condition (40–50% of cell death) avoiding high cellular mortality. Cell viability was determined by using the Cell Titer 96 Aqueous One Solution kit MTS assay (Promega, Madison, WI, USA). Specifically, 20 µL of MTS was added to each well, and incubated for 1 h at 37 °C. the absorbance was read at 492 nm using a microplate reader (Spark, Tecan, CH). Based on the MTS assay results, the concentration of 6% EtOH was chosen for the subsequent experiments. Then, the medium was replaced with 3 different concentrations of the tested NSAIDs (0.25 mM, 0.5 mM, and 1 mM) for 72 h. The results were expressed as the ratio between the absorbance of treated cells and the absorbance of CTR cells and reported as a percentage of viable cells.

### 2.4. Cell Index

Cells were seeded on a 16-well E-plate at the density of 3 × 10^5^ cell/cm^2^ cultured for 20 days post-confluence and then treated with 6% EtOH (24 h), followed by 3 different concentrations of the tested NSAIDs for 72 h (0.25 mM, 0.5 mM, and 1 mM). The dimensionless parameter cell index (CI) was evaluated using the xCELLIgence system (Roche Applied Science, DE) and used to quantify cell status based on the measured cell-electrode impedance. Normalized cell index at a certain point is obtained by dividing the CI value by the value at a reference time point. 

### 2.5. Measurement of Transepithelial Electrical Resistance 

Cells were seeded on PET membrane inserts (1.1 cm^2^ area) at a density of 3 × 10^5^ cells/cm^2^, maintained for 20 days in a complete medium, and then treated with EtOH 6% (24 h), followed by 3 different concentrations of NSAIDs (0.25 mM, 0.5 mM, and 1 mM) for 72 h. The permeability of the cellular junctions was determined by measuring the transepithelial electrical resistance (TEER) of cell monolayers using a Millicell ERS (Millipore Company, Darmstadt, Germany) employing Ag-AgCl electrodes, following the manufacturer’s protocol. The final values are expressed as Ohm*cm^2^ based on the following equation: TEER = (R − Rb) × A, where R is the resistance of filter insert with cells, Rb is the resistance of the filter alone, and A is the filter’s growth area. For subsequent experiments, the NSAID concentration of 1 mM after EtOH damage was chosen.

### 2.6. Measurement of Interleukin-8 Concentration 

The human interleukin-8 (IL-8) ELISA kit (ab46032, Abcam, Cambridge, UK) is designed for the quantitative measurement of IL-8 concentrations in the supernatants. Cells were seeded at the seeding density of 3 × 10^5^ cells/cm^2^ and maintained in a complete medium in a 100 mm dish for 20 days post-confluence. After 20 days of culture, cells were exposed to EtOH (6%) for 24 h and NSAIDs (1 mM) for 72 h. Then, the media was collected, and a gentle centrifugation was performed. For every condition, 100 µL of each sample or 100 µL of each standard were added into the appropriate well. The plate was then incubated for 1 h at room temperature. After aspirating the liquid from each well, 3 washes with 300 µL of 1X wash Buffer were performed. Then, 50 µL of 1X Biotinylated anti-IL-8 were added to each well and incubated for 1 h at room temperature. After incubation with anti-IL-8, 3 washes were performed. At this point, 100 µL of 1X Streptavidin-HRP solution was added to each well, and the plate was incubated for 30 min at room temperature. After further washes, 100 µL of chromogen 3,3′,5,5′-Tetramethylbenzidine (TMB) substrate solution was added, and the plate was incubated in the dark for 15 min at room temperature. Finally, the reaction was stopped with 100 µL of stop reagent, and the absorbance was read at 450 nm.

### 2.7. Measurement of TNF-α Concentration

A human tumor necrosis factor alpha (TNF-α) ELISA kit (ab181421, Abcam, Cambridge, UK) for the quantitative measurement of TNF-α was used. Cells were plated in a 100 mm dish (seeding density 3 × 10^5^ cells/cm^2^) in a complete medium and cultured as described above. The supernatants were collected and gently centrifuged to remove cell debris. For every condition, 50 µL of sample or standard and 50 µL of antibody cocktail were added to each well and incubated for 1 h on a plate shaker. After extensive washes, the TMB development solution was added to each well and incubated for 10 min in the dark on a plate shaker; then the reaction was stopped with the addition of 100 µL of stop solution to each well. The optical density was recorded at 450 nm.

### 2.8. Measurement of GST Activity

Glutathione-S-transferase (GST) Assay Kit (ab65326, Abcam, Cambridge, UK) is a colorimetric assay used to detect GST activity in cell lysates in order to quantify GST-tagged fusion proteins. Cells were plated (seeding density 3 × 10^5^ cells/cm^2^) into a culture-treated dish (100 mm) and cultured as described above. The EtOH 6% was added, and cells were incubated for 24 h, followed by 72 h in the different NSAID treatments upon EtOH removal. Cells were lysed in 100 µL of GST Assay Buffer for each condition and centrifuged at 10,000× *g* at 4 °C for 15 min. Supernatants were collected. In total, 25 µL of sample for all the conditions plus 25 µL of GST assay buffer was added to each well. For positive control wells, 5 µL of positive control with 45 µL of GST assay buffer was combined. At this point, 5 µL of glutathione was added to each sample and control wells, and 50 µL of the reaction mix, comprised of 49 µL of GST assay buffer and 1 µL of GST substrate solution, was added to each well. The plate was read at 340 nm in optical density.

### 2.9. Western Blotting Analysis and Protein Extraction

Protein concentration was determined with the BCA protein assay kit (Thermo Scientific, Waltham, MA, USA). This assay uses a detergent-compatible formulation based on bicinchoninic acid (BCA) for the colorimetric quantification of protein concentration. Cells were plated (seeding density 3 × 10^5^ cells/cm^2^) in a complete medium into a 100 mm dish and cultured for 20 days post-confluence. After 20 days, cells were exposed for 24 h to EtOH (6%) and for 72 h to NSAIDs (1 mM), collected and lysed in an ice-cold RIPA buffer (with freshly added protease and phosphatase inhibitors). Lysates were then diluted in sample buffer 4X (Biorad, Hercules, CA, USA). Protein samples (30–50 µg) were run in 8–12% SDS–polyacrylamide gel and electroblotted onto a polyvinyl difluoride membrane (PVDF; Sigma Aldrich, USA). Nonspecific binding sites were blocked by Every Blocking Buffer (EBB; Biorad, USA) for 5 min. Membranes were then incubated overnight at 4 °C with the following primary antibodies diluted in EBB: anti-NFκB (1:1000, ab32536, Abcam, UK), anti-IKBα (1:1000, ab32518, Abcam, UK), anti-pIKK (1:500, 2697, Cell Signaling, Danvers, MA, USA), anti-PPARγ (1:1000, ab178860, Abcam, UK), anti-4-HNE (1:1000, ab46545, Abcam, UK), anti-GSTA-4 (1:1000, ab134919, Abcam, UK), anti-Ubiquitin (1:500, GTX128826, GeneTex, Irvine, CA, USA), and Ubiquitin K48 (1:500, ab140601, Abcam, USA). As secondary antibodies, peroxidase-conjugated anti-mouse and anti-rabbit (1:20,000; Cod. 115-035-003; Cod.111-035-033; Jackson Immuno-research, Newmarket, UK) were used. Membranes were incubated with luminol (Thermo Scientific, USA) according to the manufacturer’s instructions. Bands were obtained using UVITEC (Cambridge, UK) and analyzed by ImageJ software. The relative densities of the immunoreactive bands were normalized to HRP-conjugated anti-Actin (1:10,000, 8584, Cell Signaling, MA, USA) or anti-GAPDH (1:4000, sc-32233, Santa Cruz Biotechnology, Dallas, TX, USA). Values were given as Relative Units (RU).

### 2.10. Subcellular Protein Fractionation Kit

For the subcellular protein fractionation, kit #78840, Thermo Scientific, USA was used. Caco-2 cells were plated (seeding density 3 × 10^5^ cells/cm^2^) in a complete medium into a 100 mm dish and maintained for 20 days post-confluence. After 20 days of culture, cells were exposed for 24 h to EtOH (6%) and 72 h to NSAIDs (1 mM). 

RAW 264.7 cells were seeded at 10,000 cells/cm^2^ and activated with LPS. Then, activated macrophages were treated with 1 mM of the R or S enantiomer of K or KLS for 24 h. 

For both Caco-2 and RAW 264.7 cells, the procedure of extraction of different components was performed as previously reported [18]. The protein fraction extracts were separated on a 10% SDS–polyacrylamide gel and electroblotted onto PVDF. Nonspecific binding sites were blocked by EBB for 5 min. Membranes were then incubated overnight at 4 °C with anti-NFκB and anti-PPARγ. As secondary antibodies, peroxidase-conjugated anti-rabbit, and anti-mouse were used. Bands were detected using Alliance 4.7 UVITEC (Cambridge, UK) and analyzed by ImageJ software. The relative densities of the immunoreactive bands were normalized to anti-Actin or anti-laminin using ImageJ software. Values were given as RU.

### 2.11. OxyBlot Assay Kit

OxyBlot protein oxidation detection kits (S7150, Millipore Company, Burlington, MA, USA) were used according to the manufacturer’s instructions. Briefly, cells were plated (seeding density 3 × 10^5^ cells/cm^2^) in a complete medium in a T75 flask and cultured for 20 days post-confluence. After 20 days, cells were exposed for 24 h to EtOH (6%) and for 72 h to NSAIDs (1 mM). OxyBlot assay was then performed as previously reported [19].

### 2.12. Morphological Analysis: Phalloidin iFluor 488 Staining

Cells were seeded on sterile glass coverslips at a density of 3 × 10^5^ cells/cm^2^ in a complete medium and cultured as described above. Cells were then fixed with 3.7% paraformaldehyde (Sigma, USA) in PBS for 15 min at room temperature and then permeabilized with 0.1% Triton X-100 (Sigma, USA) in PBS for 5 min. After further washes with PBS, phalloidin 1000X (ab176753, Abcam, UK) was diluted to 1X in 1% BSA in PBS, and the cells were incubated for 90 min at room temperature (according to the manufacturer’s instructions). Finally, coverslips were mounted on microscope slides with Vectashield Mounting Medium H2000 with DAPI and cells were observed using a Leica SP5 confocal microscope. For each condition, *n* = 3, 5 fields/slide were analyzed. A representative picture is shown.

### 2.13. Morphological Analysis: Immunofluorescence for ZO-1

Cells were seeded on sterile glass coverslips at a density of 3 × 10^5^ cells/cm^2^ in a complete medium and cultured for 20 days post-confluence. Cells were then treated with 6% EtOH (24 h) and then with 1 mM NSAIDs (72 h). Cells were fixed in 3.7% paraformaldehyde in PBS for 15 min at room temperature, rinsed with PBS, and permeabilized with 0.1% Triton X-100 in PBS for 5 min. After further washes with PBS, cells were incubated for 30 min with BSA 4% in PBS and then incubated overnight at 4 °C with the primary antibody zonulin-1 (ZO-1) (33-910, Thermo Scientific, MA, USA rabbit anti-human, dilution 1:100). Cells were rinsed with PBS, then incubated with secondary antibodies Alexa Fluor-488 (anti-rabbit, dilution 1:2000) and diluted in 4% BSA in PBS. Finally, coverslips were mounted on microscope slides with Vectashield Mounting Medium with DAPI and cells were observed using a Leica SP5 confocal microscope. 

### 2.14. Proteosome Trypsin-like and Caspase-like Activity Assay

Trypsin-like and caspase-like activities were determined using AMC-tagged peptide substrates (R&D system, Minneapolis, MN, USA), which release free highly fluorescent 7-amido-4-methyl coumarin (AMC) in the presence of proteasome proteolytic activity. The assay was performed in the presence or absence of bortezomib (Santa Cruz Biotechnology, Dallas, TX, USA), a selective proteasome inhibitor. Cells were seeded in T75 flasks and were exposed to EtOH for 24 h (6%), and then NSAIDS (1 mM for 72 h). Cells were then trypsinized and centrifuged for 6 min at 250× *g*. Cell pellets were rinsed with cold phosphate buffer saline, transferred into 1.5 mL tubes, then centrifuged for 6 min at 250× *g*. Then, cell pellets were resuspended in a proteasome lysis buffer and extracted as previously described [20]. Extract samples and AMC standards (1–10 μM) were placed in 96 black well plates with a final volume of 100 μL. 

In all sample wells, the fluorescent substrate AMC (final concentration 200 μM, trypsin-like) [Boc-LRR-AMC] (R&D System, MN, USA) and caspase-like [Z-LLE-AMC] (R&D System, MN, USA) was added with or without bortezomib (final concentration 100 μM). To quantify specific proteasome activity at each T (T0 or T120), the fluorescence values of the wells without inhibitors were subtracted from the fluorescence values of the wells with inhibitors. Values of proteasome activities read using a microplate reader (Spark, Tecan) correspond to the difference between fluorescence obtained in the absence of inhibitors and presence of bortezomib and the results were expressed as a percentage of CTR activity.

### 2.15. Proteosome Chymotrypsin-Like Activity Assay

Chymotrypsin activity was evaluated using a specific kit (MAK172, Sigma Aldrich, MO, USA) following the manufacturer’s protocols. Protein crude extracts were obtained and quantified as described in the previous section. Values of proteasome activities read using a microplate reader (Spark, Manneford, CH, Tecan) correspond to the difference between the fluorescence obtained in the absence of an inhibitor and the presence of bortezomib. Results were expressed in percentage of control (CTR). 

### 2.16. DCFDA Cellular Reactive Oxygen Species (ROS) Assay Kit

2′–7′-dichlorofluorescein diacetate (DCFDA, ab113851, Abcam, UK) cellular reactive oxygen species (ROS) detection assay kit was used to analyze ROS production in Caco-2 in vitro models for treatments over 48 h following manufacturer’s protocols. Cells were treated after 48 h from seeding in a 96 black well plate in a complete medium w/o phenol red (seeding density 3 × 10^5^ cell/cm^2^) and then exposed to EtOH (6%) for 24 h and then NSAIDs (1 mM for 72 h). After 72 h of NSAID exposure, cells were washed with 1X buffer and incubated with DCFDA 10 μM for 30 min at 37 °C protected from the light. H_2_O_2_ (800 μM) was used as a positive control. ROS production was instantly evaluated by determining the formation of fluorescent dichlorofluorescein at the endpoint using a microplate reader at Ex-485 nm and Em-535 nm. Data were obtained by subtracting blank readings from all measurements.

### 2.17. Crosslink Immunoprecipitation Kit

The Pierce crosslink immunoprecipitation kit (IP, 26147, Thermo Fischer, USA) allows extremely effective and efficient antigen immunoprecipitations by covalently crosslinking antibodies onto Protein A/G resin. Cells were plated (seeding density 3 × 10^5^ cells/cm^2^) in complete medium in a 100 mm dish and cultured as described above. Then, cells were collected and incubated for 5 min with 500 μL IP lysis/wash buffer for each condition. Lysates were centrifugated at 13,000× *g* for 10 min and supernatants were transferred to a new tube for protein concentration determination. The A/G plus agarose was linked with anti-NFκB (5 μg) or anti-IKBα (5 μg) for all the tested conditions according to the manufacturer’s instructions. The immunoprecipitation was performed with 1000 μg of cellular extract for all the conditions and antibodies linked to A/G plus agarose (anti-NFκB, Abcam ab32536)/anti-IKBα, Abcam ab32518). The columns were incubated overnight at 4 °C. The eluted antigen was diluted in 5X lane marker sample buffer DTT (1,4-Dithiothreitol) and then heated at 95 °C for 5 min. Eluates for NFκB or IKBα were separated on 10% SDS–polyacrylamide gel and electroblotted onto PVDF. Nonspecific binding sites were blocked with EBB for 5 min. Membranes were then incubated overnight at 4 °C with anti-ubiquitin for NFκB eluate, 4-HNE for IKBα eluate or rabbit IgG. Bands were detected using UVITEC and analyzed using ImageJ software. Values were given as RU.

### 2.18. PApp

Papp is the apparent permeability of a compound across a membrane. For drug transport studies, cells were seeded at a density of 3 × 10^5^ cells/cm^2^ on permeable support with 0.4 µm Transparent PET Membrane in complete medium, cultured and treated as described above.

Drug transport across Caco-2 monolayers was studied in the Ap–Bl direction, fresh drug and provided to the apical side (donor compartment), and the fresh vehicle was provided to the basolateral side (receiver compartment). For each solution, aliquots of 500 μL were collected immediately after the addition of compounds to the cells (the “time zero”), and after appropriate dilution, an HPLC analysis was performed. At the end of the experiment, after 120 min of incubation, 500 μL of the donor chamber solution was diluted and an HPLC analysis was performed. The apparent permeability coefficient (Papp, cm/s) of each model drug was calculated according to the following equation: Papp1/4dQ/dt/1 AC0/ð1Þ, where (dQ/dt) is the steady-state rate of the appearance of drugs in receiver side (mol/s), A is the surface area of the monolayer (cm2), and C0 is the initial compound concentration in the donor compartment (mM) (Lemieux et al., 2011).

### 2.19. Sirius T3

The major physicochemical properties of the drugs were determined using the SiriusT3 apparatus (Sirius Analytical Instruments Ltd., East Sussex, UK) equipped with an Ag/AgCl double junction reference pH electrode, a Sirius D-PAS spectrometer, and a turbidity sensing device. The pH electrode was calibrated titrimetrically in the pH range 1.8–12.2 as previously described [21]. The pKas were defined by the potentiometric method by pH-metric titration. The powder (around 0.5 mg) was diluted in 1.5 mL of ISA water and the titration was analyzed in triplicate in the pH range 2.0–12.0. Each Log P was assayed in triplicate by dissolving the powder (around 1 mg) in 1 mL of ISA water followed by pH metric titration in three different percentages of Octanol (generally 50%, 60%, 70%) [21].

### 2.20. Statistical Analyses

All data were reported as mean ± standard deviation (SD). Data analyses were performed using GraphPad Prism 8 (GraphPad Software Inc., San Diego, CA, USA). Multiple comparisons, one-way analysis of variance (ANOVA), followed by Tukey post-hoc tests were used. The level of significance was set at *p* < 0.05.

## 3. Results

### 3.1. Physicochemical Properties and Stability of Ketoprofen and Ibuprofen

In a preliminary set of experiments, the physicochemical properties pKa, logP, LogD7.4, and the thermodynamic solubility in water of K and IBU were measured using a Sirius T3 instrument. The results, together with other relevant literature data on the two NSAIDs, are reported in Table 1.

Interestingly, whereas K and IBU exhibited highly similar p*K_a_* and logP values (p*K_a_* 4.18 and 4.42 and logP 3.05 ± 0.01 and 3.91 ± 0.01, respectively), the logD7.4 (0.12) for K was lower than for IBU (logD7.4 (0.96)). Additionally, K had a significantly higher water solubility as compared to IBU (0.118 mg/mL versus 0.049 mg/mL, respectively) (Table 1). The polar surface areas of K and IBU were also compared. The higher constant surface area exposed to the GSF medium of K and KLS explains the higher intrinsic dissolution rate (IDR) in comparison to IBU, where IDR markedly influences drug absorption, distribution, metabolism, and excretion (ADME) [22]. (Table 1). 

**Table 1 cells-12-00728-t001:** Phys-chem properties and inhibitory activity on COX-1 and COX-2 of Ketoprofen and Ibuprofen.

	pKa	LogP	LogD_7.4_	Solubility	PSA	IC_50_ COX-1 (Human Blood)	IC_50_ COX-2 (Human Blood)
Ketoprofen	4.18	3.05 ± 0.01	0.12	118 µg/mL	75.7 Å^2^	0.047 µM [23] 0.11 µM [24]	2.9 µM [23] 0.88 µM [24]
Ibuprofen	4.42	3.91 ± 0.01	0.96	49 µg/mL	49.1 Å^2^	7.6 µM [23] 5.9 µM [24]	7.2 µM [23] 9.9 µM [24]

Our results combined with previous work suggest that KLS exhibits greater solvation kinetics, thus showing a better permeability profile than other agents tested [25].

### 3.2. Cell Viability and Cell Index

Caco-2 cells treated with EtOH were used as a model of leaky gut. To fix the model, Caco-2 cells were cultured for 48 h and then exposed to different concentrations of EtOH (2%, 4%, 6%, and 8%) to find a sub-toxic condition (40–50% of cell death), such to avoid high cellular mortality (Appendix A). Based on the MTS assay results, 6% EtOH was selected for the subsequent experiments. In parallel, cells were cultured for 20 DIV to create a condition that closely resembles the in vivo barrier, treated with 6% EtOH and then exposed to three different concentrations of IBU, IBU-Lys, IBU-Arg, K, KLS, Arg, and Lys (0.25 mM, 0.5 mM, and 1 mM) for 72 h. All the compounds tested significantly affected cell viability with the exception of KLS, showing a significant increase of cell viability with respect to ETOH treatment.

The health of the Caco-2 epithelial cells was evaluated by CI (Figure 1A) and TEER assays (Figure 1B). Both analyses showed a protective effect exerted by KLS in the epithelial recovery phase following EtOH damage with a full recovery of injury in the TEER assay. Interestingly, in the TEER assay, IBU showed a partial but significant protective activity on EtOH-induced damage that was not potentiated in the Lys salt form (Figure 1B). Based on cell viability results, a concentration of 1 mM was selected for the subsequent experiments (Appendix A).

### 3.3. Morphological Analysis of the Intestinal Barrier

The intestinal barrier integrity was evaluated by the immunostaining for ZO-1 protein (Figure 2A) and phalloidin staining of actin microfilaments (Figure 2B). Upon EtOH challenge, both ZO-1 and phalloidin expression are strongly reduced, indicating an evident disruption of the barrier layer. Among the items tested, only KLS treatment protected the monolayer morphology, partially preserving the barrier integrity (Figure 2A,B). Seeing that K and Lys alone did not show the ability to protect Caco-2 cells from the EtOH-induced insult, we investigated the mechanistic basis of the synergic-like effect observed in cells exposed to KLS. 

### 3.4. Proinflammatory Signals

Increased intestinal barrier permeability leads to inappropriate proinflammatory signaling that then further contributes to gut barrier dysfunction [26,27]. To characterize these effects in our model, we first evaluated the influence of EtOH on the stimulation of proinflammatory cytokines, such as IL-8 and TNF-α. Secreted IL-8 and TNF-α levels (Figure 3A,B) were significantly increased by EtOH exposure compared to control (283.9 ± 4.1 pg/mL vs. 82.4 ± 0.44 pg/mL and 20.9 ± 0.7 pg/mL vs. 10.8 ± 1.3 pg/mL, respectively), while all K or IBU and their corresponding Lys and Arg salts were able to counteract this effect, thus supporting the hypothesis that COX activation is a key event in EtOH-induced amplification of inflammatory signals. 

The effect of the EtOH challenge on NFκB, a key regulator of inflammatory processes [28], was investigated. In non-stimulated cells, NFκB complexes are sequestered in the cytoplasm in an inactive form by interaction with the monomeric form of the inhibitory IKB protein [29]. The NFkB/IKB system is finely regulated by a complex network of events in response to different stimuli, such as LPS, oxidative stress, or proinflammatory cytokines. Under stimulation, the IKB cytoplasmatic inhibitor (IKK) complex phosphorylates IKBα leading to its ubiquitination and proteasomal degradation, triggering nuclear translocation of NFkB and gene expression induction [30]. EtOH treatment induced the nuclear translocation of NFκB, accompanied by a marked reduction of IKBα protein levels and a concomitant increase in the protein levels of pIKK, the active form of IKK (Figure 3C and Figure 4A,B). Interestingly, only K and KLS treatments significantly counteracted all these effects, modulating the stress induced by NFkB/IKB pathway activation. Consistent with the observed effect on NFκB translocation, IKBα protein levels, measured by western blotting analysis, were significantly reduced upon the EtOH challenge as compared to the control, and only K and KLS counteracted the EtOH-induced IKBα decrease (Figure 4A). 

We also tested the effect of EtOH on the regulation of PPARγ, a ligand-dependent nuclear receptor whose activation results in the inhibition of NFκB signaling and inflammatory cytokine production [31,32]. EtOH treatment reduced the level of PPARγ in the nucleus, thus triggering NFκB activation. Notably, K and KLS treatments restored the nuclear translocation of PPARγ almost completely (Figure 4C). 

### 3.5. Oxidative Stress and Proteasome Activity

In previous studies on the gastric mucosa damage model [17], we demonstrated the effect of KLS in decreasing oxidative stress by downregulating HNE protein adducts, and hypothesized that the ε-amino group of Lys may directly react with oxidant aldehydes including 4-HNE—a terminal product of lipid peroxidation—and other oxidant aldehydes. Thus, we tested the hypothesis that a direct antioxidant action of Lys could account for the observed synergism. The obtained results refuted our hypothesis by showing that only KLS, and not Lys, was able to reduce the elevation of 4-HNE levels caused by EtOH (Figure 5A). IKBα/4-HNE adduct formation was also evaluated by immunoprecipitation for IKBα (Figure 5B). EtOH increased protein levels for IKBα/4-HNE adducts and, in agreement with the above results, only K, and KLS with higher efficacy—but not Lys—reduced the formation of the adducts.

Having excluded a direct aldehyde scavenging effect of Lys, we evaluated its action on key cellular detoxification mechanisms. Glutathione-S-transferase (GST) is a family of enzymes that exert a key role in the detoxification of xenobiotics [33]. In Figure 5C, the quantitative analysis of GST activity is shown. As expected, GST activity was significantly decreased by the EtOH challenge, while only the treatment with KLS counteracted this effect that was not observed with both K and LYS alone. It is known that glutathione S-transferase A4 (GSTA4) reduces intracellular levels of 4-HNE [34]. In agreement, in Figure 5D, EtOH treatment strongly reduced GSTA-4 protein levels, while only K and KLS treatments counteracted the HNE-adducts formation with a more pronounced effect of KLS. Due to the evidence of oxidative stress occurring upon the EtOH challenge, an OxyBlot assay was used to determine the levels of oxidized protein. High levels of oxidized proteins, due to non-physiological oxidative stress levels, are related to increased intracellular proteins damaged by oxygen free radicals. Our results confirmed that Lys did not exhibit direct antioxidant effects in this cellular model but significantly enhanced the antioxidant effect of K. This effect may be due to the ability of Lys to interfere with transport through the intestinal epithelium as reported previously [35,36], thus favouring K permeability.

Starting from the observation (Figure 4) that K exhibits a specific modulatory effect on the NFkB pathway-specific and independent-from-COX inhibitory activity, we designed additional studies to further investigate biological significance and mechanisms. The levels of oxidized proteins were increased by EtOH damage, while K, and more significantly KLS, counteracted this effect (Figure 5E). IBU and related salts did not show a protective effect, suggesting a COX-independent mechanism underlying the observed reduction of oxidized proteins levels. On the other hand, the DCFDA assay (an indicator of cellular ROS) showed a significant reduction in fluorescence intensity in all the tested compounds, which was more substantial upon K and KLS treatment, compared to the increased intensity due to the EtOH challenge alone (Figure 5F).

Oxidative stress and oxidized protein overload affect proteasome activity. Polyubiquitin chains linked via lysine 48 (K48) are the most abundant and represent the canonical signal for protein degradation by the proteasome. EtOH-induced damage increased levels of ubiquitin K48 and ubiquitin (Figure 6A,B), an effect markedly counteracted by KLS treatment. Immunoprecipitation for ubiquitinated NFκB complex (IKB-NFκB) was evaluated by WB. The EtOH challenge increased the levels of ubiquitinated NFκB/IKB complex, inducing the degradation of IKB and the nuclear translocation of NFκB. Consistent with the above results, both K and KLS were able to counteract this effect (Figure 6C). Finally, EtOH treatment strongly reduced all the proteasome activities tested, but this effect was attenuated upon KLS treatment only (Figure 6D–F). 

### 3.6. Effects of R and S Ketoprofen Enantiomers on Activated Macrophages

Having found that K has a specific activity in the inhibition of the inflammatory NFkB pathway that cannot be explained by its known inhibitory activity on COX enzymes since it is not exhibited by IBU, we hypothesized that this outcome may be a previously unreported effect of the R-enantiomer of K. The R-enantiomer of K—several logs less potent than the S enantiomer as a COX inhibitor [37,38]—was previously reported to contribute to the in vivo efficacy of the racemic drug by exerting analgesic and anti-inflammatory effects through a COX-independent mechanism [39]. We tested the effects of the two enantiomers of K, R, and S, on NFkB activation. Interestingly, only R and not the S enantiomer inhibited NFkB translocation to the same extent as racemic K and KLS (Figure 7A,B).

To further corroborate our hypothesis, we tested R and S enantiomers on activated macrophages. We found that only R and not the S enantiomer inhibited NFkB translocation to the same extent as KLS, suggesting that the R enantiomer contributes to the in vivo anti-inflammatory effect of K and KLS (Figure 7C,D). 

### 3.7. Permeability of R and S ketoprofen in Caco2 Cells

Furthermore, to assess the human intestinal permeability of R and S enantiomers through the epithelial barrier, we performed a permeability assay in Caco-2 cells. The results showed low permeability from the apical to basolateral direction. Interestingly, we observed that KLS showed a higher permeability compared to K (Figure 7E). We further evaluated the permeability of the single enantiomer and we found no differences in terms of their Papp (K: Papp = 0.579 ± 0.052 10^−6^ cm/s for R enantiomer and 0.576 ± 0.031 10^−6^ cm/s for S enantiomer; KLS: Papp = 1.185 ± 0.044 10^−6^ cm/s for R enantiomer and 1.186 ± 0.026 10^−6^ cm/s for S enantiomer) (Figure 7E).

## 4. Discussion

NSAIDs injure both the gastric and intestinal mucosa and are related to numerous gastrointestinal (GI) problems, including relatively mild, nuisance-type symptoms such as heartburn, nausea, dyspepsia, and abdominal discomfort [40,41]. However, in as many as 80% of NSAIDs users, acute haemorrhages and mucosal erosions are identified in the gastroduodenal mucosa upon endoscopic examination [42,43], which can induce severe consequences including bleeding and perforation, erosions, ulcers, strictures, and bowel obstruction. Elderly and chronic NSAID users are especially affected by NSAID-associated side effects. NSAID management is expected to increase to relieve aging-related degenerative and inflammatory disorders; at the same time, the occurrence of NSAID-associated side effects is also likely to rise. As long-term users suffer from diminished absorption capacity and increased intestinal permeability, they are at risk for developing colitis and exacerbating pre-existing conditions such as IBD and irritable bowel syndrome (IBS) [43].

The mechanism of NSAID-induced gastric injury is still unclear. TJs participate in cell-cell adhesion and are present in epithelial and endothelial cell membranes, establishing a component of intercellular junctions and exerting crucial functions in barrier integrity, cell polarity, and cell signalling pathways [44]. TJs generate a paracellular barrier that can be altered when NSAIDs harm the gastric epithelium, leading to enhanced permeability [45]. The loss of barrier integrity is a characteristic feature of different conditions, including microbiota dysbiosis, obesity, and IBD [46]. A leaky gut, or dysfunction of the intestinal barrier increasing permeability [47], may be an early occurrence in the pathogenesis of many GI disorders, permitting bacteria-derived molecules to enter into the mucosa and leading to uncontrollable inflammatory signal cascades [47]. 

Among other reagents, EtOH can induce leaky gut and intestinal epithelial barrier (IEB) dysfunction [48] that the use of NSAIDs may exacerbate. Here, we used the EtOH challenge as a model of intestinal barrier injury. We found that EtOH exposure induced a complete loss of IEB function, as shown by the results of CI and TEER assays and morphological analysis. Testing the effects of K, IBU, and their corresponding Lys and Arg salts on cells stressed by EtOH allowed us to observe that only KLS rescued IEB function following EtOH injury, thus showing a clear KLS advantage versus both K and IBU or IBU-Lys and IBU-Arg.

We compared the effect of IBU, K, and their salts on the stress-induced oxidative pathways in the IEB. In line with our previous observations on the gastric mucosa, KLS decreased ROS and HNE formation, supporting a model by which K, and more potently KLS, reduces the formation of HNE protein adducts by triggering proteasome activity that leads to their degradation via GST increase. IBU and IBU-Lys did not show any modulation of these parameters suggesting COX-independent effects. In the EtOH-induced IEB damage model, a general decrease of oxidized proteins and oxidative stress was observed upon KLS treatment, suggesting that proteasome activity was rescued. We previously reported that the accumulation of oxidized proteins and protein adducts triggers proteasomal overload resulting in proteasome dysfunction [20]. The results here reported the marked protective effects of KLS already reported in the stomach mucosa [20] extending to the IEB, increasing knowledge on the involved signalling pathways. 

Starting from these preliminary observations, in this work, we tried to investigate the mechanisms by which: (a) K, differently from IBU, stimulates the IEB protective maintenance in response to stress, and (b) Lys potentiates the protective effect of K treatment. 

In our previous work, we formulated the hypothesis of a direct scavenging activity by Lys on cytotoxic reactive aldehydes such as HNE. The results reported here confirm the ability of Lys to favour the preservation of proteasome activity by decreasing oxidative stress response, decreasing oxidized protein accumulation, preserving the functionality of key enzymes involved in oxidative stress control, and preserving the regulation of anti-inflammatory and protective pathways, like PPARγ or ZO-1. Our results, not showing any inhibitory effect exerted by Lys alone, did not support the direct scavenging effect hypothesis, undoubtedly confirming an evident synergic contribution of the Lys counterion on the specific properties of KLS. 

An alternative possible interpretation of these observations is related to the ability of Lys to alter the permeability and intracellular distribution of K, as apparently suggested by the results of the studies on Caco-2 [35,36] in which KLS showed a higher permeability compared to K. A Lys-mediated permeability increase was highly specific since IBU-Lys did not exhibit different permeability vs. IBU, and this could be explained by the ability of Lys to interfere with the cationic amino acid channels expressed in the basolateral membrane of intestinal epithelial cells [35,36] that regulate the Na + Lys trafficking and may indirectly influence the transport system involved in K absorption and excretion.

The results so far generated in both gastric mucosa and IEB damage models do not provide a full clarification of the mechanisms by which Lys potentiates the positive effect of K on IEB injury repair and warrants future studies aimed at elucidating molecular mechanisms underlying the observed modulation of inflammatory and oxidative response. 

On the other hand, the results of this study nicely contribute to outlining the mechanism responsible for the specific COX-independent mechanism observed for K in the EtOH-induced stress IEB model. In fact, whereas K, IBU, and their salts showed similar behaviour on stress-induced cytokine production, a specific effect of K was observed in the modulation of the NFkB/PPARγ pathways.

EtOH treatment significantly increased the nuclear translocation of NFkB and decreased the nuclear translocation of PPARγ. These two transcription factors are known to act in opposite directions, the first inducing inflammatory signals and oxidative stress, and the latter counteracting inflammation and ROS formation [49,50,51]. In our in vitro model, K was able to increase nuclear PPARγ and decrease nuclear NFkB in a COX-independent manner. The K effect was potentiated by the presence of Lys in KLS. In this context, it is worth noting that K is as racemic formulation, and to gain more insight into K-specific effects in the NFkB pathway, the R and S enantiomers were checked. This assay allowed us to demonstrate for the first time that the R enantiomer, which is the less potent COX inhibitor [52], is fully responsible for the observed effects on NFkB translocation and activity. This finding not only offers a potential mechanistic explanation about the specific effect of K in the protection and repair of the damaged intestinal epithelial barrier, but also sheds new light on the mechanisms by which the R-enantiomer may contribute to the anti-inflammatory and analgesic effect of KLS. The specific effect of R-Ketoprofen on the NFkB pathway may also explain its reported COX-independent effect in pain management, as NFkB upregulation was recently reported in spinal and glial cells [38] as a key mediator of hyperalgesia in several animal models. Additionally, the observation that R-enantiomer inhibits NFkB nuclear localization and activation in human macrophages supports this hypothesis and prompts additional in vitro and in vivo studies. Our results suggest that the activity of R-enantiomers should be considered for molecules administered as racemic mixtures. This result agrees with a previous Phase I randomized study comparing the gastroduodenal side effects of similar doses of racemic ketoprofen and (R) and (S)-ketoprofen on normal subjects, showing that the number of gastric lesions induced by (S)-ketoprofen and racemic ketoprofen was found to be double the number of lesions observed after(R)-ketoprofen [53]. 

Since COX1 and COX2 inhibitory characteristics fail to fully explain the observed differences between various NSAIDs, many resources have been dedicated to identifying novel targets that may account for the specific properties. Aspirin was the first of the NSAIDs that was shown to suppress NF-kB [54], and others were subsequently characterized for the same effect [55,56,57,58]. That has been proposed as crucial for the comprehension of NSAID’s specific pharmacological characteristics, in particular regarding pain management. In this sense, NFkB upregulation has been recently suggested in spinal and glial cells as a key mediator of hyperalgesia in several experimental models [59,60,61].

The results reported here show for the first time a specific effect of R-Ketoprofen on the NFkB pathway that sheds new light on the previously reported [38] COX-independent effect in pain management, and that may account for the observed unexpected protective effect of K on stress-induced damage on the IEB. The cooperative effect of R-Ketoprofen and Lys counterion accounts for the unique efficacy and tolerability characteristics of KSL within the class of commercially available NSAIDs. The possible synergic activity of R-ketoprofen and lysine, and the effect of the lysine salt of the R-stereoisomer versus the lysine salt of the S-stereoisomer will be investigated further in future in vitro and in vivo studies, to provide the rationale for introducing the R-KLS into clinical practice. 

## Figures and Tables

**Figure 1 cells-12-00728-f001:**
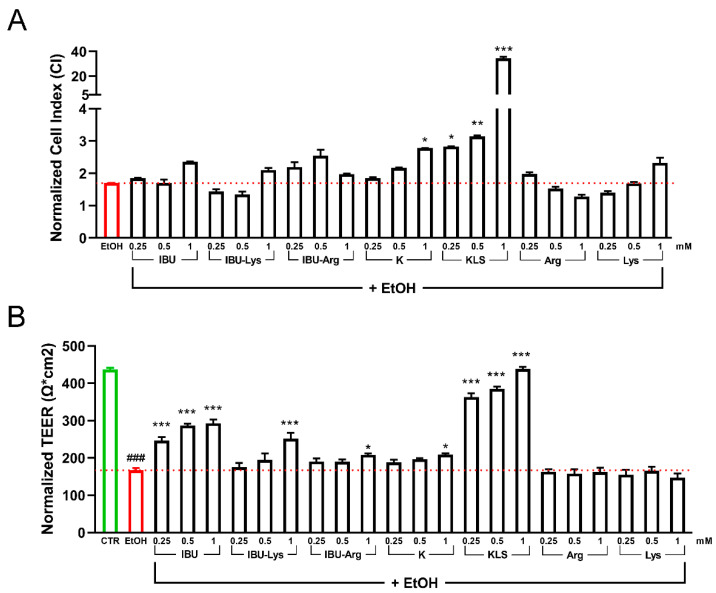
(**A**) Cell index analysis for Caco-2 cultured 20 DIV, treated with 6% EtOH for 24 h and then with the tested NSAIDs and corresponding salts (0.25 mM, 0.5 mM, and 1 mM for 72 h). Data are reported as Normalized Cell index. (**B**) TEER analysis for Caco-2 cultured for 20 DIV, treated with EtOH 6% for 24 h, and then with the tested NSAIDs (0.25 mM, 0.5 mM, and 1 mM for 72 h). Data are reported as Normalized TEER. Data are mean ± SD of 3 different experiments. One way ANOVA was performed: ***, *p* < 0.0001; **, *p* < 0.005; *, *p* < 0.05 vs. EtOH; ###, *p* < 0.0001 EtOH vs. CTR.

**Figure 2 cells-12-00728-f002:**
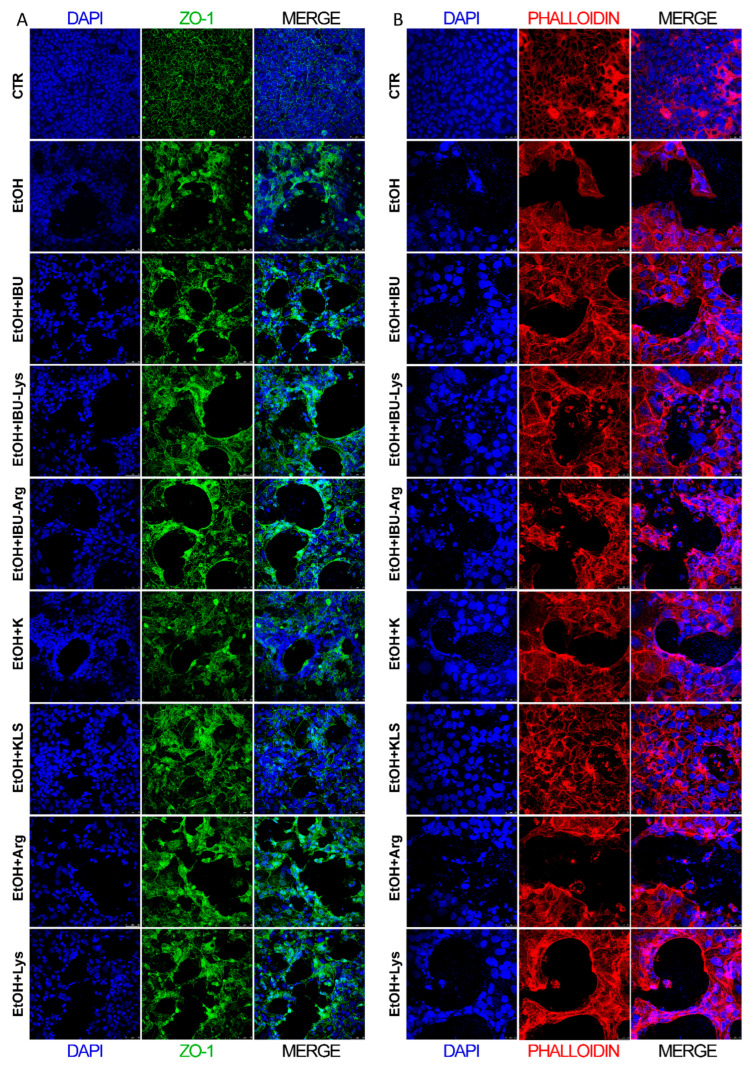
(**A**) Immunofluorescence of ZO-1 in Caco-2 cells after EtOH (6% for 24 h), EtOH + IBU, EtOH + IBU-Lys, EtOH + IBU-Arg, EtOH + K, EtOH + KLS, EtOH + Arg, and EtOH + Lys (1 mM) treatments for 72 h. ZO-1 (green), DAPI (nuclei, blue) Bar = 50 µm. (**B**) Immunofluorescence of phalloidin (red) staining in Caco-2 cells after EtOH (6% for 24 h), EtOH + IBU, EtOH + IBU-Lys, EtOH + IBU-Arg, EtOH + K, EtOH + KLS, EtOH + Arg, and EtOH + Lys (1 mM) treatments for 72 h. DAPI (nuclei, blue). Bar = 25 µm. *n* = 3, 5 fields/slide were examined, and a representative picture is shown.

**Figure 3 cells-12-00728-f003:**
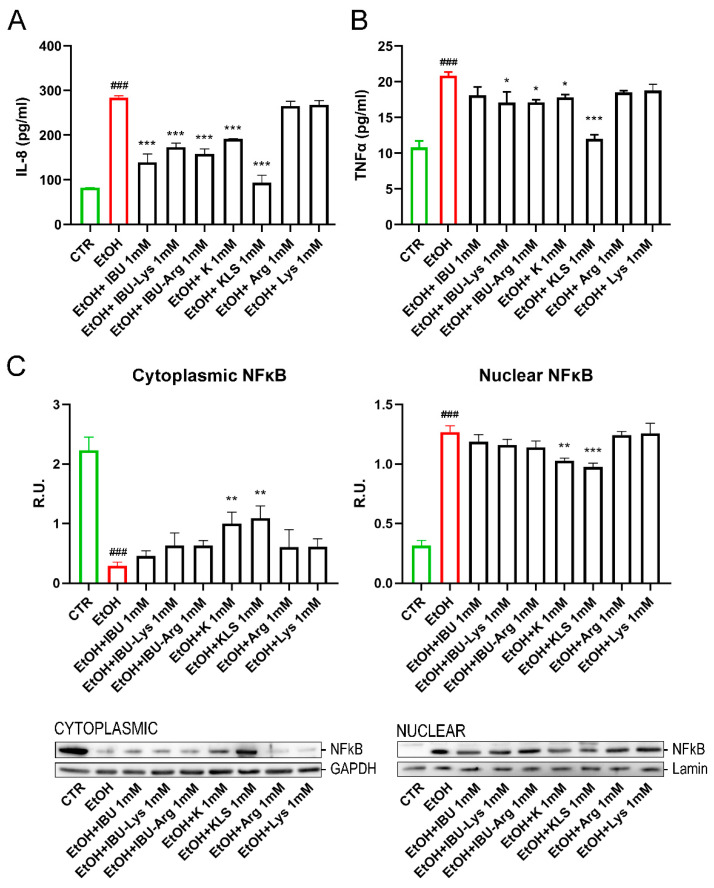
(**A**) Secreted levels of IL-8 on Caco-2 cells after EtOH (6% for 24 h), EtOH + IBU, EtOH + IBU-Lys, EtOH + IBU-Arg, EtOH + K, EtOH + KLS, EtOH + Arg, and EtOH + Lys treatments (1 mM) for 72 h by ELISA assay. (**B**) Secreted levels of TNFα on Caco-2 cells after EtOH (6%, for 24 h), EtOH + IBU, EtOH + IBU-Lys, EtOH + IBU-Arg, EtOH + K, EtOH + KLS, EtOH + Arg, and EtOH + Lys treatments (1 mM) treatments for 72 h by ELISA assay. Data are mean ± SD of 3 different experiments. (**C**) WB analysis for cytoplasmic and nuclear components of NFκB on Caco-2 cells after EtOH (6%, for 24 h), EtOH + IBU, EtOH + IBU-Lys, EtOH + IBU-Arg, EtOH + K, EtOH + KLS, EtOH + Arg, and EtOH + Lys treatments (1 mM) for 72 h. Data are mean ± SD of 3 different experiments. One way ANOVA was performed: ***, *p* < 0.0001; **, *p* < 0.005; *, *p* < 0.05 vs. EtOH; ###, *p* < 0.0001 EtOH vs. CTR.

**Figure 4 cells-12-00728-f004:**
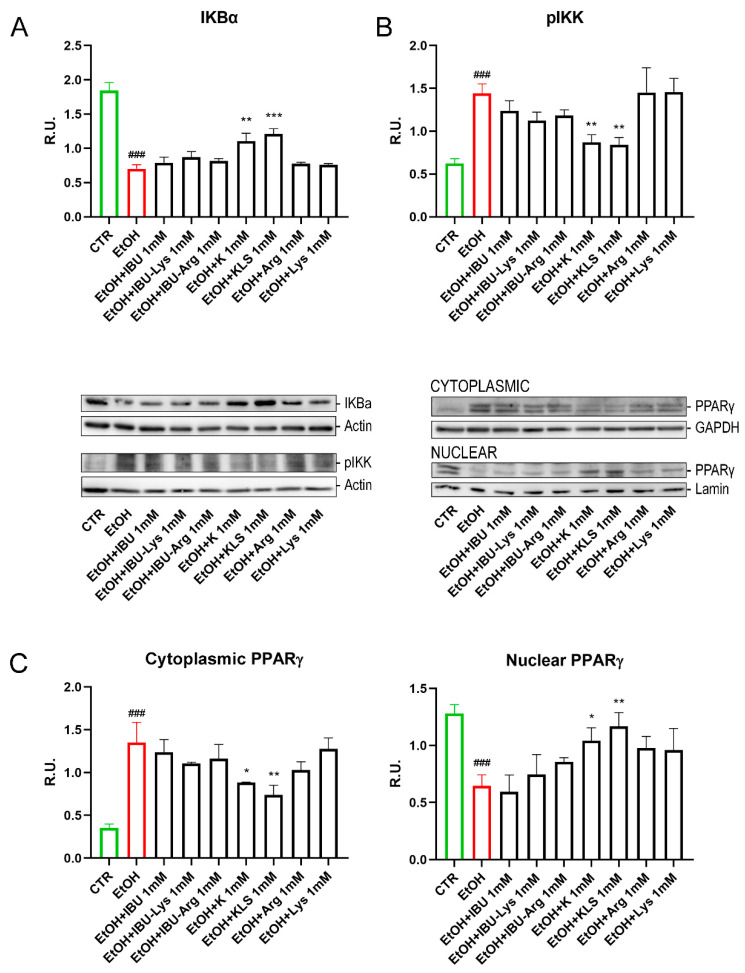
(**A**) WB analysis for IKBα (cytoplasmic and nuclear components) on Caco-2 cells after EtOH (6%, for 24 h), EtOH + IBU, EtOH + IBU-Lys, EtOH + IBU-Arg, EtOH + K, EtOH + KLS, EtOH + Arg, and EtOH + Lys treatments (1 mM) for 72 h. (**B**) WB analysis for the active form of IKK (pIKK) (cytoplasmic and nuclear components) on Caco-2 cells after EtOH (6%, for 24 h), EtOH + IBU, EtOH + IBU-Lys, EtOH + IBU-Arg, EtOH + K, EtOH + KLS, EtOH + Arg, and EtOH + Lys treatments (1 mM) for 72 h. Representative images of WB for IKBα, pIKK and PPARγ (cytoplasmic and nuclear components) are shown. (**C**) WB analysis for PPARγ (cytoplasmic and nuclear components) on Caco-2 cells after EtOH (6% for 24 h), EtOH + IBU, EtOH + IBU-Lys, EtOH + IBU-Arg, EtOH + K, EtOH + KLS, EtOH + Arg, and EtOH + Lys treatments at 1 mM for 72 h. Data are mean ± SD of 3 different experiments. One way ANOVA was performed: ***, *p* < 0.0001; **, *p* < 0.005; *, *p* < 0.05; vs. EtOH; ###, *p* < 0.0001 EtOH vs. CTR.

**Figure 5 cells-12-00728-f005:**
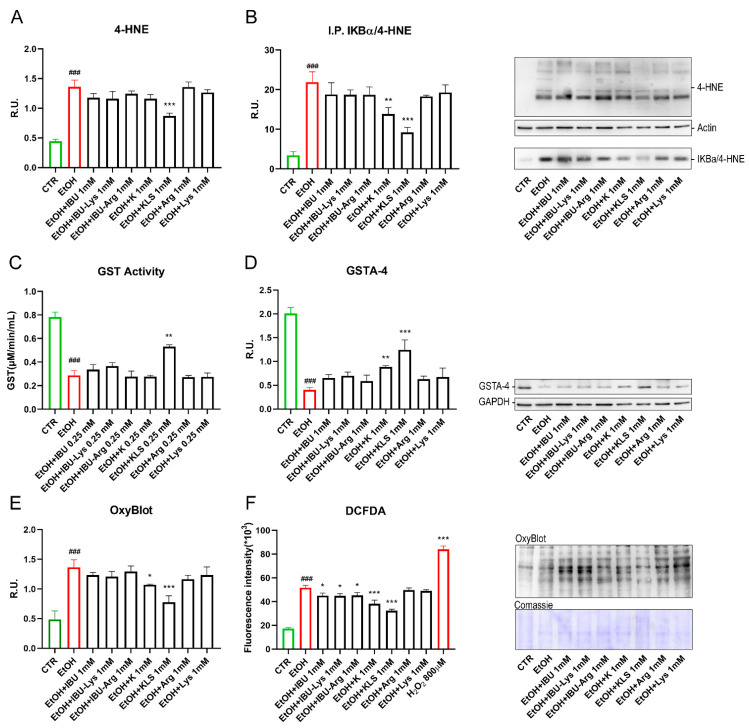
(**A**) WB analysis for 4-HNE on Caco-2 cells after EtOH (6% for 24 h), EtOH + IBU, EtOH + IBU-Lys, EtOH + IBU-Arg, EtOH + K, EtOH + KLS, EtOH + Arg, and EtOH + Lys treatments (1 mM for 72 h). Data are mean ± SD of 3 different experiments. (**B**) Immunoprecipitation for IKBα on Caco-2 cells after EtOH, EtOH + IBU, EtOH + IBU-Lys, EtOH + IBU-Arg, EtOH + K, EtOH + KLS, EtOH + Arg, and EtOH + Lys treatments (1 mM for 72 h). A representative western blot image is reported. (**C**) GST activity analyzed by ELISA assay on Caco-2 cells after EtOH (6% for 24 h), EtOH + IBU, EtOH + IBU-Lys, EtOH + IBU-Arg, EtOH + K, EtOH + KLS, EtOH + Arg, and EtOH + Lys treatments (1 mM for 72 h). (**D**) WB analysis for GSTA-4 on Caco-2 cells after EtOH (6% for 24 h), EtOH + IBU, EtOH + IBU-Lys, EtOH + IBU-Arg, EtOH + K, EtOH + KLS, EtOH + Arg, and EtOH + Lys treatments (1 mM for 72 h). A representative western blot image is reported. (**E**) OxyBlot assay to assess the intracellular oxidized protein on Caco-2 cells after EtOH (6% for 24 h), EtOH + IBU, EtOH + IBU-Lys, EtOH + IBU-Arg, EtOH + K, EtOH + KLS, EtOH + Arg, and EtOH + Lys treatments (1 mM for 72 h). A representative OxyBlot image is reported. (**F**) DCFDA assay of Caco-2 upon EtOH (6% for 24 h), EtOH + IBU, EtOH + IBU-Lys, EtOH + IBU-Arg, EtOH + K, EtOH + KLS, EtOH + Arg and EtOH + Lys treatments. Data are mean ± SD of 3 different experiments. One way ANOVA was performed: ***, *p* < 0.0001; **, *p* < 0.005; *, *p* < 0.05 vs. EtOH; ###, *p* < 0.0001 EtOH vs. CTR.

**Figure 6 cells-12-00728-f006:**
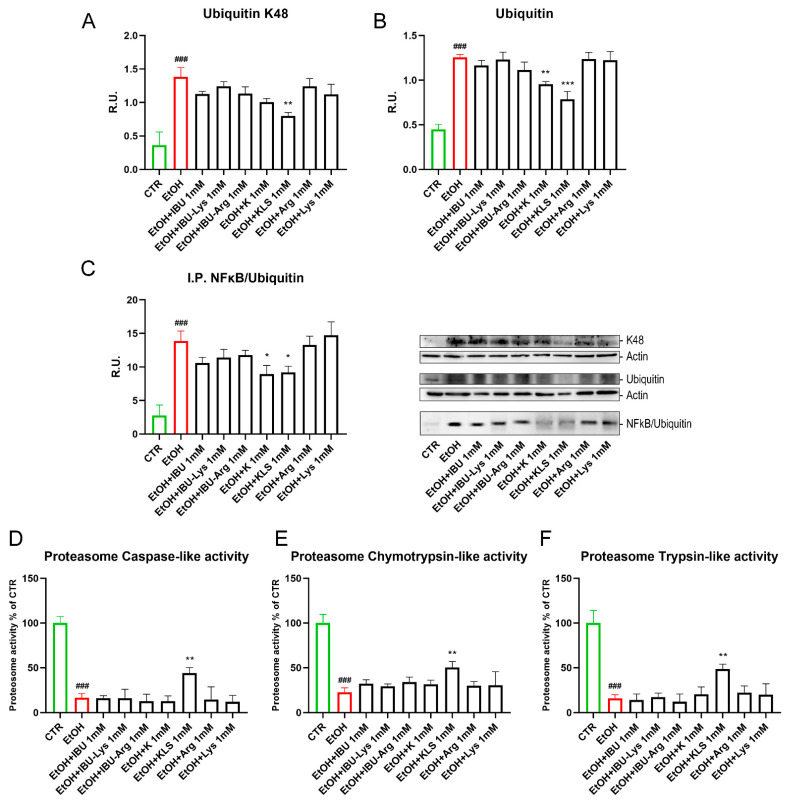
(**A**) WB analysis for ubiquitin K48 and (**B**) Ubiquitin on Caco-2 cells after EtOH challenge (6%, for 24 h), EtOH + IBU, EtOH + IBU-Lys, EtOH + IBU-Arg, EtOH + K, EtOH + KLS, EtOH + Arg, and EtOH + Lys treatments (1 mM for 72 h). (**C**) Immunoprecipitation for NFκB on Caco-2 cells after EtOH challenge (6%, for 24 h), EtOH + IBU, EtOH + IBU-Lys, EtOH + IBU-Arg, EtOH + K, EtOH + KLS, EtOH + Arg, and EtOH + Lys treatments (1 mM for 72 h). Data are mean ± SD of 3 different experiments. The image blot is a representative one. (**D**–**F**) Proteasome activities on Caco-2 cells after EtOH and NSAID treatments. Data are mean ± SD of 3 different experiments. One way ANOVA was performed: ***, *p* < 0.0001; **, *p* < 0.005; *, *p* < 0.05 vs. EtOH; ###, *p* < 0.0001 EtOH vs. CTR.

**Figure 7 cells-12-00728-f007:**
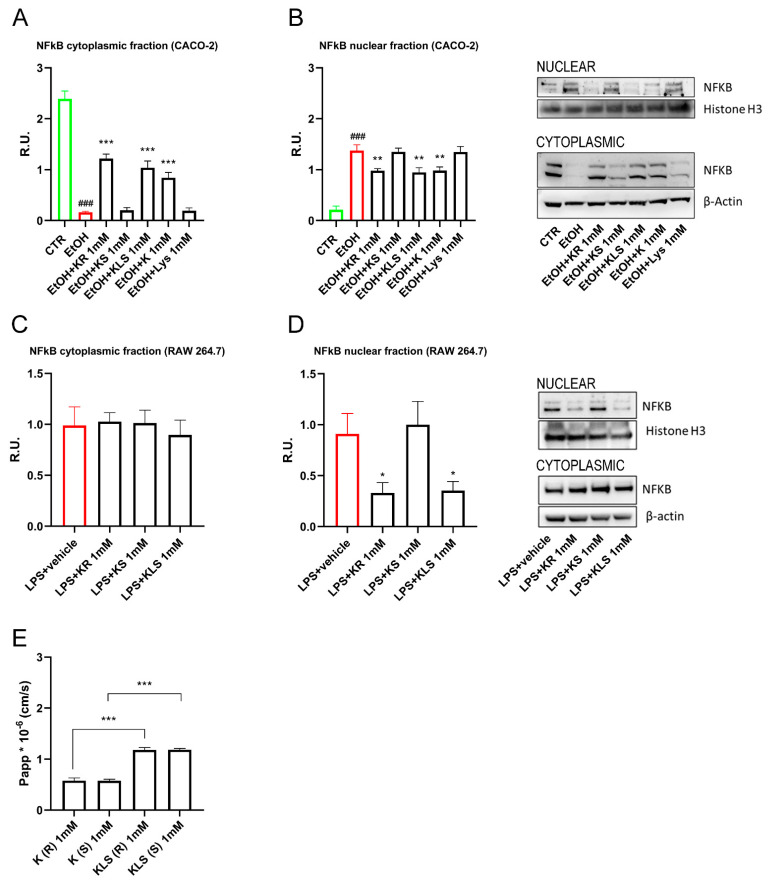
WB analysis for cytoplasmatic and nuclear components of NFκB (**A**,**B**) Caco-2 cells and (**C**,**D**) LPS-activated RAW 264.7 upon R or S enantiomer of K and K, KLS, Lys for 24 h. One way ANOVA was performed: ***, *p* < 0.0001; **, *p* < 0.005; *, *p* < 0.05; vs. EtOH; ###, *p* < 0.0001 EtOH vs. CTR. One-way ANOVA: * *p* < 0.05 vs. LPS vehicle (CTR) *n* = 3. The image blot is a representative one. (**E**) Permeability assay of Caco-2 cells upon R or S enantiomer of K, KLS. One-way ANOVA was performed: ***, *p* < 0.0001; **, *p* < 0.005 vs. EtOH. One-way ANOVA: * *p* < 0.05 vs. LPS vehicle (CTR) *n* = 3.

## Data Availability

Raw data can be provided by Correspond Authors upon reasonable request.

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
