# Peer review of "Differential Effects of Nonsteroidal Anti-Inflammatory Drugs in an In Vitro Model of Human Leaky Gut"

_cells, 2023, doi:10.3390/cells12050728_

Round 1

Reviewer 1 Report

In the article by d’Angelo et al (entitled: “Differential Effects of Nonsteroidal Anti-Inflammatory Drugs on an in vitro model of human leaky gut”) the authors provide several lines of evidence that ketoprofen and/or its lysine salt reverse pathological characteristics associated with the leaky gut syndrome, using an in vitro model where cells are exposed to ethanol.

I have several suggestions for the improvement of the manuscript, as well as a few additional points that need further discussion.

Title

I suggest a slight correction of the article title (word change from “on” to “in”): “Differential effects of Nonsteroidal Anti-Inflammatory Drugs in an in vitro model of human leaky gut”.

Abstract

The abstract needs to be rewritten. In its current form the abstract only provides background information that served as the basis of the study. There is no information regarding the methods used and main results. The conclusion is also not adequate for the results obtained in the study – the authors suggest potential relevance of their results for pain management, whereas their study focused on the effects of ketoprofen on intestinal epithelium.

Experimental design, methods and results

Examination of physicochemical properties: How were data for PSA obtained? Were they experimentally derived or do they represent literature data?

Choice of NSAID concentrations: Initially the authors examined the effects of 3 different concentrations of ibuprofen, ketoprofen and their salts (0.25, 0.50 and 1.00 mM), and in their further experiments they focused on the 1.00 mM concentration. It would be beneficial for the authors to discuss if these concentrations are of potential relevance for in vivo effects, i.e. can these high concentrations be achieved when ketoprofen or its salts are therapeutically used?

Cell viability assay:  The results from the MTS assay (which are presented in the supplemental figure) are somewhat confusing for me. Based on the presented results it can be concluded that all tested compounds (with the exception of the lysine salt of ketoprofen) reduce cell viability, however this is not explicitly discussed in the manuscript. In addition, the authors should include an appropriate legend for the supplemental figure.

Morphological analysis: Immunofluorescence experiments appear to have been performed only on one sample per treatment. Why have the authors not performed experiments on more samples and why is there no quantitative analysis of the intensity of immunofluorescence (and statistical analysis of differences between treatments)?

GST activity: Based on the presented results, it would appear that ibuprofen, its arginine salt, lysine and arginine significantly further reduce GST activity. The authors should re-check the results of their statistical analysis, and discuss how these compounds further reduce GST activity.

PAPP: The authors should expand the methods section regarding the PAPP analysis. What does the abbreviation PAPP stand for? What is the purpose of this experiment and how were this experiments conducted (provide detail or references to the method that was used).

Figure 7 (labelling mistake): There is a mistake in the labelling of blots in figure 7 – cytoplasmatic blots for NFKB are wrongly labelled as nuclear blots and vice versa.

Statistical analysis

The authors performed a relatively basic and non-complex form of statistical analysis. However, the significance of the results seems to be apparent. I would suggest that the authors expand the reported results of their statistical analysis and provide p values for non-significant results, as well se p values for comparisons between ketoprofen and ketoprofen-lysine salt (in some cases there appears to be a statistical superiority of the lysine salt compared to ketoprofen).

Introduction and Discussion

One of the most notable findings of this study is that ketoprofen, a well-known NSAID, as well as its lysine salt exerted protective effects against damage of epithelium cells induced by ethanol. The presented results are, in my opinion, surprising, considering that ketoprofen is an NSAID known to have a high potential to cause damage to the gastrointestinal tract (for reference, see: Castellsague et al. Individual NSAIDs and upper gastrointestinal complications: a systematic review and meta-analysis of observational studies (the SOS project). Drug Saf. 2012 Dec 1;35(12):1127-46). The authors should discuss how their results align with real-life data indicating the unfavorable gastrointestinal safety profile of ketoprofen. Additionally, the protective effects were seen with relatively high concentrations of ketoprofen – are these concentrations relevant for the potential therapeutic use of ketoprofen? (see text above regarding the choice of NSAID concentrations)

Furthermore, the authors link the protective effects of ketoprofen to its R-enantiomer. The authors could expand their discussion and include published clinical data regarding the better safety profile of R-ketoprofen compared to racemic ketoprofen (see: Jerussi et al. Clinical endoscopic evaluation of the gastroduodenal tolerance to (R)- ketoprofen, (R)- flurbiprofen, racemic ketoprofen, and paracetamol: a randomized, single-blind, placebo-controlled trial. J Clin Pharmacol. 1998 Feb;38(2S):19S-24S).

Author Response

In the article by d’Angelo et al (entitled: “Differential Effects of Nonsteroidal Anti-Inflammatory Drugs on an in vitro model of human leaky gut”) the authors provide several lines of evidence that ketoprofen and/or its lysine salt reverse pathological characteristics associated with the leaky gut syndrome, using an in vitro model where cells are exposed to ethanol.

I have several suggestions for the improvement of the manuscript, as well as a few additional points that need further discussion.

Title

I suggest a slight correction of the article title (word change from “on” to “in”): “Differential effects of Nonsteroidal Anti-Inflammatory Drugs in an in vitro model of human leaky gut”.

We agree with the reviewer and changed the Title as suggested.

Abstract

The abstract needs to be rewritten. In its current form the abstract only provides background information that served as the basis of the study. There is no information regarding the methods used and main results. The conclusion is also not adequate for the results obtained in the study – the authors suggest potential relevance of their results for pain management, whereas their study focused on the effects of ketoprofen on intestinal epithelium.

We agree with the reviewer, the Abstract is now modified.

Experimental design, methods and results

Examination of physicochemical properties: How were data for PSA obtained? Were they experimentally derived or do they represent literature data?

The PSA, Polar Surface Area, was calculated with ACD Percepta Software (ACD Lab Release 2019.2.2)

Choice of NSAID concentrations: Initially the authors examined the effects of 3 different concentrations of ibuprofen, ketoprofen and their salts (0.25, 0.50 and 1.00 mM), and in their further experiments they focused on the 1.00 mM concentration. It would be beneficial for the authors to discuss if these concentrations are of potential relevance for in vivo effects, i.e. can these high concentrations be achieved when ketoprofen or its salts are therapeutically used?

Yes, concentration levels of about 1 mM can be reached locally at intestinal level. In fact, after oral therapeutic dose, the Cmax of ketoprofen is in the range 10-23 mg/l which is approximately equivalent to a plasma concentration of 0.1 mM.  Since the compound has a high intestinal adsorption, it is reasonable to assume that after oral administration, local intestinal concentration reaches concentration levels of about 1 mM.

Cell viability assay:  The results from the MTS assay (which are presented in the supplemental figure) are somewhat confusing for me. Based on the presented results it can be concluded that all tested compounds (with the exception of the lysine salt of ketoprofen) reduce cell viability, however this is not explicitly discussed in the manuscript. In addition, the authors should include an appropriate legend for the supplemental figure.

We now better illustrate the suppl Figure and introduce a legend for this figure

Morphological analysis: Immunofluorescence experiments appear to have been performed only on one sample per treatment. Why have the authors not performed experiments on more samples and why is there no quantitative analysis of the intensity of immunofluorescence (and statistical analysis of differences between treatments)?

The experimental procedure is not well explained, for each condition, three slides were processed. For each slide, 5 field/slide, was examined. However, the analysis is, in this case, not quantitative but qualitative since it investigated the presence/damage of the cellular junctions and not their amount.

GST activity: Based on the presented results, it would appear that ibuprofen, its arginine salt, lysine and arginine significantly further reduce GST activity. The authors should re-check the results of their statistical analysis, and discuss how these compounds further reduce GST activity.

We appreciate the reviewer’s comment and we totally agree. We apologized for oversight since we noticed that the text and statistical significance were correct, whereas the figure was incorrect. We now replaced with the proper figure.

PAPP: The authors should expand the methods section regarding the PAPP analysis. What does the abbreviation PAPP stand for? What is the purpose of this experiment and how were this experiments conducted (provide detail or references to the method that was used).

The reviewer is right, we did not explain well the Papp calculation method. Papp is the apparent permeability of a compound across a membrane. It Is a parameter used for determining drug transport. For drug transport studies, cells were seeded on ThinCert multiwell plates (Greiner Bio-One GmbH Frickenhausen,Germany). Drug transport across Caco monolayers was studied the Ap–Bl direction, fresh drug and provided to the apical side (donor compartment) and the fresh vehicle was provided to the basolateral side (receiver compartment). For each solution, aliquots of 500 μL were collected immediately after the addition of compounds to the cells (the "time zero") and after appropriate dilution, HPLC analysis was performed. At the end of the experiment, after 120 minutes of incubation, 500 μL of the donor chamber solution was diluted and HPLC analysis was performed. The apparent permeability coefficient (Papp, cm/s) of each model drug was calculated according to the following equation: Papp1/4dQ/dt/1 AC0/ð1Þ, where (dQ/dt) is the steady-state rate of appearance of drugs in receiver side (mol/s), A is the surface area of the monolayer (cm2), and C0 is the initial compound concentration in the donor compartment (mM) (Lemieux et al., 2011). We introduced now this description in the Matherial Method Section.

Figure 7 (labelling mistake): There is a mistake in the labelling of blots in figure 7 – cytoplasmatic blots for NFKB are wrongly labelled as nuclear blots and vice versa.

We apologize for the mistake, we now corrected the figure.

Statistical analysis

The authors performed a relatively basic and non-complex form of statistical analysis. However, the significance of the results seems to be apparent. I would suggest that the authors expand the reported results of their statistical analysis and provide p values for non-significant results, as well se p values for comparisons between ketoprofen and ketoprofen-lysine salt (in some cases there appears to be a statistical superiority of the lysine salt compared to ketoprofen).

We thank the reviewer for the comment and we now added a file reporting p values.

Introduction and Discussion

One of the most notable findings of this study is that ketoprofen, a well-known NSAID, as well as its lysine salt exerted protective effects against damage of epithelium cells induced by ethanol. The presented results are, in my opinion, surprising, considering that ketoprofen is an NSAID known to have a high potential to cause damage to the gastrointestinal tract (for reference, see: Castellsague et al. Individual NSAIDs and upper gastrointestinal complications: a systematic review and meta-analysis of observational studies (the SOS project). Drug Saf. 2012 Dec 1;35(12):1127-46). The authors should discuss how their results align with real-life data indicating the unfavorable gastrointestinal safety profile of ketoprofen. Additionally, the protective effects were seen with relatively high concentrations of ketoprofen – are these concentrations relevant for the potential therapeutic use of ketoprofen? (see text above regarding the choice of NSAID concentrations)

We agree with the reviewer, our work clearly points out the need of prospective comparative studies in order to better assess the relative risk ratio of different drugs belonging to the NSAIDs class. We acknowledge that meta-analysis of observational studies suggests counterintuitive data on the comparative safety profile, in particular between K and IBu. The limitation of these results is associated with the different real-world use of the two molecules in as much as ibuprofen is largely the most used compound in the OTC market at the strength of 200 mg/Kg. The equivalent strength of Ketoprofen is 25 mg/Kg that has been introduced in the OTC market after 2015 and only in a limited number of countries. Before that time most of the data on Ketoprofen are referred to the higher strengths used in the RX market (from 50 to 20o or more) that should be compared to Ibuprofen 600 mg.  The review dates to 2012 but additional studies will help in a final clinical confirmation and evaluation of the significance of our in vitro finding. 

Regarding the relatively high concentration used, it should be noted that following oral administration in humans the Cmax of ketoprofen is in the range 10-23 mg/l which is approximately equivalent to a plasma concentration of 0.1 mM.  Since the compound has a high intestinal adsorption and to be close to a concentration that is not too far from a toxic one, we decided to test the 1 mM.

Furthermore, the authors link the protective effects of ketoprofen to its R-enantiomer. The authors could expand their discussion and include published clinical data regarding the better safety profile of R-ketoprofen compared to racemic ketoprofen (see: Jerussi et al. Clinical endoscopic evaluation of the gastroduodenal tolerance to (R)- ketoprofen, (R)- flurbiprofen, racemic ketoprofen, and paracetamol: a randomized, single-blind, placebo-controlled trial. J Clin Pharmacol. 1998 Feb;38(2S):19S-24S).

We now included the clinical data suggested and included the suggested references.

Reviewer 2 Report

The submitted article (Differential Effects of Nonsteroidal Anti-Inflammatory Drugs on an in vitro model of human leaky gut) describes the activity of two NSAIDs, ketoprofen and ibuprofen, and their salts with Arg and Lys on several biochemical parameters related to inflammation and the undesired effects of NSAIDs on the gastrointestinal tract, in vitro.

I think that this paper could be published in Cells only after a thorough revision. In particular:

1) The authors should make clear from the beginning that the used NSAIDs are racemic mixtures. Furthermore, if they are racemates, then their salt with Arg and Lys (considering that the L-aminoacids are used, which must also be clarified), are diastereomers, i.e. two different compounds for each NSAID, with different physical, and perhaps chemical properties. In fact, the authors use in each experiment two compounds for every drug, which makes the interpretation of the results ambiguous. The conditions of the preparation, isolation and identification of all salts and their biological activities have to be presented.

2) There are not any data for KLS in Table 1. The authors state that “Our results combined with previous work suggest that KLS exhibits greater solvation kinetics, thus showing a better permeability profile than other agents tested [24]” l. 325-326. However, ref. 24 seems irrelevant.

3) In the abstract the authors mention that “The present study aims to compare the effects of distinct classes of NSAIDs, such as ketoprofen (K), Ibuprofen (IBU), and their corresponding lysine (Lys) and arginine (Arg) salts” ) l. 42-44, however this stands only for ibuprofen, and only one, the Lys salt of ketoprofen, is studied.

4) Authors should decide how to spell NF-kappaB.

Author Response

The submitted article (Differential Effects of Nonsteroidal Anti-Inflammatory Drugs on an in vitro model of human leaky gut) describes the activity of two NSAIDs, ketoprofen and ibuprofen, and their salts with Arg and Lys on several biochemical parameters related to inflammation and the undesired effects of NSAIDs on the gastrointestinal tract, in vitro.

I think that this paper could be published in Cells only after a thorough revision. In particular:

1) The authors should make clear from the beginning that the used NSAIDs are racemic mixtures. Furthermore, if they are racemates, then their salt with Arg and Lys (considering that the L-aminoacids are used, which must also be clarified), are diastereomers, i.e. two different compounds for each NSAID, with different physical, and perhaps chemical properties. In fact, the authors use in each experiment two compounds for every drug, which makes the interpretation of the results ambiguous. The conditions of the preparation, isolation and identification of all salts and their biological activities have to be presented.

We have used the racemic forms as commercial drugs have NSAIDs in racemic form. The racemic forms were compared to each other as reported in previous studies [ref 17]. The compounds were prepared starting from the chiral commercially available starting material and according to the procedure described in [New resolving bases for ibuprofen and mandelic acid: qualification by binary phase diagrams. J.E. Ebbers; M.B. Plum, J.A. Gerry; A. Ariaans; B. Kaptein; Q.B. Broxterman; A. Bruggink; B. Zwanenburg. Tetrahedron Asymmetry Vol 8, 4047 – 4057, 1997.] This information are now included in the ms.

2) There are not any data for KLS in Table 1. The authors state that “Our results combined with previous work suggest that KLS exhibits greater solvation kinetics, thus showing a better permeability profile than other agents tested [24]” l. 325-326. However, ref. 24 seems irrelevant.

We thank the reviewer for the correction and we apologize for the uncorrected reference. We corrected reference number 24.  [Pain and ketoprofen: what is its role in clinical practice?; P. Sarzo-Puttini; F. Atzeni; L. Lanata; M. Bagnasco; M. Colombo; F. Fischer; M. D’Imporzano. Reumatismo, 2010; 62(3):172-188]

3) In the abstract the authors mention that “The present study aims to compare the effects of distinct classes of NSAIDs, such as ketoprofen (K), Ibuprofen (IBU), and their corresponding lysine (Lys) and arginine (Arg) salts” ) l. 42-44, however this stands only for ibuprofen, and only one, the Lys salt of ketoprofen, is studied.  

The ketoprofen arginine salt was not considered in this study and this is now corrected in the abstract and in the introduction. This is due to the fact, that ketoarg is not presently used on humans and it is only present for veterinary use.

4) Authors should decide how to spell NF-kappaB.

We thank the reviewer we have now correct NFkB through the ms.

Reviewer 3 Report

This is a well-done study performed by an experienced Team of scientists, who have largely contributed and published on the topic. Although their experimental approach to the subject has been sound in the past, compared to previous work, this investigation does represent a step forward. The results obtained are original and interesting and deserve priority publication.

The translational value of this work is highly relevant since NSAIDs are widely used worldwide. Inflammation with its symptoms, signs and functional impairment is indeed encountered in everyday clinical practice by any specialist physician. While no one will doubt about NSAID efficacy, concerns about safety do represent an issue. Amongst the many adverse effects, those related to the GI tract are the most common and sometimes life-threatening. However, while NSAID-gastropathy is nowadays well appreciated by practicing physicians, the GI damage beyond the duodenum (whose symptoms and signs are subtle) is often overlooked. In both upper and lower NSAID-induced GI injury the increase in mucosal permeability does represent a primum movens that sets in motion a series of pathophysiologic events leading to gross lesion formation.

I have no major criticism to this work. However, in the hope to increase the already high level of the paper, I would like to give some (hopefully useful) suggestions.

General Comments

On lines 106-114, they mention the different combinations of compounds tested. However, the effect of ketoprofen and ibuprofen (as well as their lysine and arginine salts) alone is not shown. I am sure the Authors should have these data in their database.

While the data presented clearly show the protective effect of KLS on EtOH-induced changes, it would be interesting – especially from a translational point of view – to understand the effect of the different NSAIDs per se. Indeed, although in clinical practice NSAIDs may be given in patients taking alcoholic beverages (which do increase drug-induced injury), this is not customary. However, NSAIDs may be given to patients (like those with Inflammatory Bowel Disease, Celiac Disease, Irritable Bowel Syndrome, Rheumatoid Arthritics, etc.), in whom an underlying increase of intestinal permeability is present. Therefore, the effect of these drugs on EtOH-induced changes (as a model of leaky gut) is relevant from a clinical standpoint.

Due to the possible synergic activity of R-ketoprofen and lysine, the effect of the lysine salt of the R-stereoisomer versus the lysine salt of the S-stereoisomer should be investigated in vitro and in vivo. Provided the present results be confirmed, they would represent the rationale for introducing the R-KLS into clinical practice. I believe this important consideration should be included at the end of the DISCUSSION section.

Specific Comments

Although the paper is globally well written, it is not always fluent and easy to follow. I would suggest that – in the preparation of a revised version – the Authors will ask the help of a scientist, who is a native speaker.

While the METHODS section is well organized, this is not the case for the RESULTS section. I would suggest to adopt – in the presentation of data – the same framework, that is to divide the section in sub-sections, related to each kind of investigation.

Minor Comments

On line 68, “…… causes of intestinal permeability….” should be “…… causes of increased intestinal permeability….”

On line 98, “….to stimulate leaky gut…..” should be “….to induce leaky gut…..”

On line 105, the Authors mention the “Experimental conditions”. I would suggest to write “Different sets of in vitro experiments were performed”.

On lines 136, 144, 156, 163, etc. it’s better to write “Measurement of Transepithelial Electrical Resistance” instead of “Transepithelial Electrical Resistance assay”. Along the same lines, instead of “Interleukin-8 ELISA Assay Kit”, “Measurement of Interleakin-8 Concentration” would be preferable

Author Response

This is a well-done study performed by an experienced Team of scientists, who have largely contributed and published on the topic. Although their experimental approach to the subject has been sound in the past, compared to previous work, this investigation does represent a step forward. The results obtained are original and interesting and deserve priority publication.

The translational value of this work is highly relevant since NSAIDs are widely used worldwide. Inflammation with its symptoms, signs and functional impairment is indeed encountered in everyday clinical practice by any specialist physician. While no one will doubt about NSAID efficacy, concerns about safety do represent an issue. Amongst the many adverse effects, those related to the GI tract are the most common and sometimes life-threatening. However, while NSAID-gastropathy is nowadays well appreciated by practicing physicians, the GI damage beyond the duodenum (whose symptoms and signs are subtle) is often overlooked. In both upper and lower NSAID-induced GI injury the increase in mucosal permeability does represent a primum movens that sets in motion a series of pathophysiologic events leading to gross lesion formation.

I have no major criticism to this work. However, in the hope to increase the already high level of the paper, I would like to give some (hopefully useful) suggestions.

 We really thank the reviewer for the appreciation, we tried to meet all the points raised.

General Comments

On lines 106-114, they mention the different combinations of compounds tested. However, the effect of ketoprofen and ibuprofen (as well as their lysine and arginine salts) alone is not shown. I am sure the Authors should have these data in their database.

The different compounds were used alone and the MTS assay was performed. This information is now included as a new suppl Fig. 2

While the data presented clearly show the protective effect of KLS on EtOH-induced changes, it would be interesting – especially from a translational point of view – to understand the effect of the different NSAIDs per se. Indeed, although in clinical practice NSAIDs may be given in patients taking alcoholic beverages (which do increase drug-induced injury), this is not customary. However, NSAIDs may be given to patients (like those with Inflammatory Bowel Disease, Celiac Disease, Irritable Bowel Syndrome, Rheumatoid Arthritics, etc.), in whom an underlying increase of intestinal permeability is present. Therefore, the effect of these drugs on EtOH-induced changes (as a model of leaky gut) is relevant from a clinical standpoint.

 As outlined in the first part of the work, for the purpose of this work we have selected the maximal concentration of the drugs devoid of direct toxic mediated effect in order to dissect the EtOH-induced toxicity mechanism from concurrent toxicity associated to COX-inhibition in order to specifically study the effect on leaky-gut conditions.

Due to the possible synergic activity of R-ketoprofen and lysine, the effect of the lysine salt of the R-stereoisomer versus the lysine salt of the S-stereoisomer should be investigated in vitro and in vivo. Provided the present results be confirmed, they would represent the rationale for introducing the R-KLS into clinical practice. I believe this important consideration should be included at the end of the DISCUSSION section.

Thank for the thoughtful comment. The direct comparison between S-enantiomer and the recemic mixture will be investigated in a subsequent work under finalisation.

This point is now stressed in the Discussion section 

Specific Comments

Although the paper is globally well written, it is not always fluent and easy to follow. I would suggest that – in the preparation of a revised version – the Authors will ask the help of a scientist, who is a native speaker.

The ms was now revised by a native English Speaker. 

While the METHODS section is well organized, this is not the case for the RESULTS section. I would suggest to adopt – in the presentation of data – the same framework, that is to divide the section in sub-sections, related to each kind of investigation.

 We now modified accordingly the reviewer’s suggestion

Minor Comments

On line 68, “…… causes of intestinal permeability….” should be “…… causes of increased intestinal permeability….”

Corrected 

On line 98, “….to stimulate leaky gut…..” should be “….to induce leaky gut…..”

 Corrected

On line 105, the Authors mention the “Experimental conditions”. I would suggest to write “Different sets of in vitro experiments were performed”.

Corrected according to the reviewer

On lines 136, 144, 156, 163, etc. it’s better to write “Measurement of Transepithelial Electrical Resistance” instead of “Transepithelial Electrical Resistance assay”. Along the same lines, instead of “Interleukin-8 ELISA Assay Kit”, “Measurement of Interleukin-8 Concentration” would be preferable

Corrected.